# Structural basis of ribosomal peptide macrocyclization in plants

**Joel Haywood[1,2], Jason W Schmidberger[1,2], Amy M James[1,2], Samuel G Nonis[1,2], Kirill V Sukhoverkov[1,2], Mikael Elias[3], Charles S Bond[1], Joshua S Mylne[1,2]***

[1]School of Molecular Sciences, The University of Western Australia, Perth, Australia; [2]The ARC Centre of Excellence in Plant Energy Biology, University of Western Australia, Crawley, Australia; [3]Department of Biochemistry, Molecular Biology and Biophysics, University of Minnesota, Minneapolis, United States

**Abstract** Constrained, cyclic peptides encoded by plant genes represent a new generation of drug leads. Evolution has repeatedly recruited the Cys-protease asparaginyl endopeptidase (AEP) to perform their head-to-tail ligation. These macrocyclization reactions use the substrates amino terminus instead of water to deacylate, so a peptide bond is formed. How solvent-exposed plant AEPs macrocyclize is poorly understood. Here we present the crystal structure of an active plant AEP from the common sunflower, *Helianthus annuus*. The active site contained electron density for a tetrahedral intermediate with partial occupancy that predicted a binding mode for peptide macrocyclization. By substituting catalytic residues we could alter the ratio of cyclic to acyclic products. Moreover, we showed AEPs from other species lacking cyclic peptides can perform macrocyclization under favorable pH conditions. This structural characterization of AEP presents a logical framework for engineering superior enzymes that generate macrocyclic peptide drug leads.
DOI: https://doi.org/10.7554/eLife.32955.001

## Introduction

Asparaginyl endopeptidases (AEPs) are a group of asparagine/aspartic acid (Asx) specific proteases that have been classified as belonging to the C13 family of cysteine proteases based on the presence of a His-Gly-spacer-Ala-Cys motif (*Hara-Nishimura et al., 1993*; *Chen et al., 1997*; *Mathieu et al., 2002*; *Shafee et al., 2015*). First described in plants as vacuolar processing enzymes based on their propensity for processing seed proteins stored in vacuoles, AEPs have since been described in a variety of organisms and shown to be involved in a wide range of processes including, cell death antigen processing and hemoglobin degradation (*Hara-Nishimura et al., 1993*; *Manoury et al., 1998*; *Hatsugai et al., 2004*; *Kuroyanagi et al., 2005*; *Yamada et al., 2005*; *Sojka et al., 2007*). In addition to the proteolytic function observed in these processes, AEP has become well known for its curious ligation reactions (*Min and Jones, 1994*; *Sheldon et al., 1996*; *Mylne et al., 2012*; *Nguyen et al., 2014*; *Zhao et al., 2014*; *Dall et al., 2015*).

The ability of endoproteases to perform ligation reactions was first observed by Bergmann and Fruton in 1938 with chymotrypsin (*Bergmann and Fruton, 1938*). Later, in vitro ligation reactions were performed with AEP from jack bean (*Canavalia ensiformis*) seeds (*Bowles et al., 1986*; *Min and Jones, 1994*). The recent discovery that evolutionarily distinct plant families have repeatedly recruited AEPs to catalyze the formation of ribosomally synthesized and post-translationally modified peptides (RiPPs), through the macrocyclization of linear precursor sequences, has caught the attention of drug designers keen to overcome the current inefficiencies in native chemical ligation that limit the therapeutic use of cyclic peptides (*Pattabiraman and Bode, 2011*; *Mylne et al., 2012*; *Arnison et al., 2013*). Such therapeutic cyclic peptides are viewed by many to have the potential to capitalize on a niche in the current pharmaceutical market by virtue of their intermediate size

**\*For correspondence:**
joshua.mylne@uwa.edu.au

**Competing interests:** The authors declare that no competing interests exist.

**eLife digest** Most proteins are long, chain-like molecules that have two ends respectively called the N-terminus and C-terminus. However, certain proteins can close on themselves to become circular. This requires a chemical reaction between the N- and C-termini, which creates a strong bond between the two extremities.

To go through this 'cyclization' process, a straight protein attaches to a certain type of protease, a class of enzyme that usually cuts proteins into smaller pieces. In plants that are distantly related, the same group of enzymes – called AEPs – has been selected to perform cyclization. Here, Haywood et al. study an AEP enzyme from sunflowers: they identify what about this enzyme's structure is important to drive the complex chemical reaction that results in the protein being cyclized rather than simply cut.

Using a technique called X-ray crystallography to see the positions of individual atoms in the enzyme, Haywood et al. caught a snapshot of the enzyme. Its structure explained how the enzyme's shape can guide cyclization. In particular, the part of the enzyme that binds to the proteins, the active site, was relatively flat and open, but also flexible: this helped the N and C-termini react with each other and close the protein. Further experiments artificially mutated specific areas of the enzyme, which helped determine exactly which elements guide this succession of chemical reactions.

The activity of AEPs is influenced by their local environment, such as acidity. In fact, Haywood et al. showed that certain AEPs, which do not normally carry out cyclization, can start performing this role when exposed to a different level of acidity.

The pharmaceutical industry is increasingly interested in circular proteins, as these are stable, easily used by the body, and can be genetically customized to act only on specific targets. If the cyclization process is better understood, and then harnessed, new drug compounds could be produced.

DOI: https://doi.org/10.7554/eLife.32955.002

between small molecule drugs and large protein structures, and their unique capacity to combine favorable bioavailability and stability characteristics with high target specificity facilitated by tolerance to site-directed mutagenesis (*Clark et al., 2005*, *2010*; *Gould et al., 2011*; *Ji et al., 2013*; *Poth et al., 2013*; *Truman, 2016*). Moreover, as computational techniques for the discovery of RiPPs improve and the number of cyclic peptides described continues to expand, an ever-wider array of scaffolds might be exploited to tailor molecules to specific drug targets (*Bhardwaj et al., 2016*; *Truman, 2016*; *Hetrick and van der Donk, 2017*) (*Figure 1*).

Sunflower trypsin inhibitor-1 (SFTI-1) is a 14-residue, bicyclic peptide with a cyclic backbone and an internal disulfide bond (*Luckett et al., 1999*). Its biosynthesis is rather unusual as its sequence is buried within a precursor that also encodes seed storage albumin. Seed storage albumins are a major class of seed storage protein that constitute over 50% of total seed protein and become a source of nitrogen and sulfur during seed germination (*Youle and Huang, 1978*; *Shewry and Halford, 2002*). The common sunflower (*Helianthus annuus*) has many genes encoding precursors for these napin-type or 2S seed storage albumins that are synthesized in the rough endoplasmic reticulum before undergoing cleavage maturation by AEP and localizing to storage vacuoles (*Bollini and Chrispeels, 1979*; *Franke et al., 2016*; *Jayasena et al., 2016*). Along with an adjacent albumin, SFTI-1 is post-translationally processed by AEP from within a unique seed storage albumin precursor called Preproalbumin with SFTI-1 (PawS1) (*Mylne et al., 2011*). SFTI-1 is a potent inhibitor of serine proteases, and its intrinsic stability and cellular penetration capabilities have led to its application as a bioactive scaffold (*White and Craik, 2016*; *Swedberg et al., 2017*).

AEP-catalyzed macrocyclization of SFTI-1 is hypothesized to proceed via a cleavage-coupled intramolecular transpeptidation reaction whereby catalysis begins with the deprotonation of the active site Cys by a localized His of the catalytic center, facilitating the nucleophilic attack on the carbonyl carbon of the Asp by the activated Cys thiol of AEP (*Bernath-Levin et al., 2015*). This attack culminates in the formation of a thioacyl intermediate between the substrate and AEP, and the removal of the C-terminus. The reaction is proposed to be subsequently concluded by the nucleophilic attack of this intermediate by a Gly at the N-terminus of the substrate, resulting in the

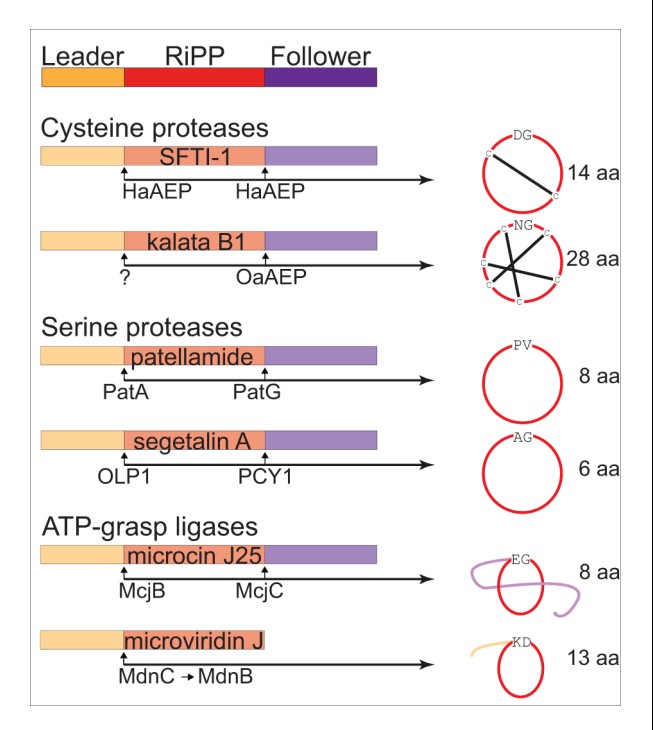

**Figure 1.** Examples of enzyme catalyzed formation of cyclic RiPPs. Cyclic RiPPs that are enzyme catalyzed from linear peptide precursors are commonly flanked by an N-terminus leader sequence and a C-terminus follower sequence prior to cyclization. These flanking sequences commonly aid in substrate recognition and catalysis. Well characterized examples of enzymes that are able to perform macrocyclization currently include cysteine proteases (*Helianthus annuus AEP* – HaAEP, and *Oldenlandia affinis* AEP – OaAEP; following removal of the leader peptide by an as yet undefined enzyme), serine proteases (patellamide A/G – PatA/G, and peptide cyclase 1 – PCY1; following removal of the leader peptide by oligopeptidase 1 – OLP1) and the ATP dependent ATP-grasp ligases (microviridin C and B – MdnC/B, and microcin J25 C - McjC; following removal of the leader peptide by microcin J25 B – McjB). These examples also illustrate a range of cyclic scaffolds that may have potential therapeutic applications.

DOI: https://doi.org/10.7554/eLife.32955.003

macrocyclization of SFTI-1 in a head-to-tail manner (*Mylne et al., 2011*; *Bernath-Levin et al., 2015*). Notably, this reaction proceeds in competition with nucleophilic attack upon the thioacyl intermediate by any nearby water molecules which would produce hydrolyzed, acyclic-SFTI. In vitro studies revealed the ratio of acyclic to cyclic SFTI-1 to be in the order of 5.8:1, with the less stable acyclic products hypothesized to be quickly degraded in vivo (*Bernath-Levin et al., 2015*). Evidence for AEP-mediated and hydrolysis-independent transpeptidation was demonstrated through the exclusion of a heavy atom $O^{18}$ in the cyclic SFTI-1 product from an in vitro jack bean AEP (CeAEP1) catalyzed reaction (*Bernath-Levin et al., 2015*). The formation of a much larger cyclic peptide by an AEP from *Oldenlandia affinis* (OaAEP1) was also shown to lack $O^{18}$ incorporation, suggesting a conserved mechanism of macrocyclization despite differences in substrate sequences (*Harris et al., 2015*). However, the suggestion that an AEP from butterfly pea (*Clitoria ternatea*) termed butelase one functions only as a ligase (*Nguyen et al., 2014*), combined with the proposal of a succinimide-driven, cleavage-independent ligation event based on the crystal structure of human AEP (hAEP) and its ability to ligate substrate in the absence of the catalytic Cys, has cast uncertainty on the mechanism of AEP-catalyzed macrocyclization (*Dall et al., 2015*).

In order to elucidate a structural explanation why plant AEPs have been recruited by distinct plant lineages to perform macrocyclization and to understand the catalytic and structural nuances that might allow preferences towards cleavage or ligation reactions, we sought the crystal structure of a sunflower AEP.

Herein, we describe the first structure of an active plant AEP; one capable of performing peptide macrocyclization. This AEP, the most abundant AEP of five AEPs in the common sunflower (HaAEP1), displays structural similarity to previously published active AEPs from mammals, with subtle differences at residues involved in substrate recognition. Our characterization by site-directed mutagenesis of HaAEP1 residues integral to macrocyclization will facilitate the bioengineering of plant AEPs for improved macrocyclization efficiency, diversifying the scaffolds usable as cyclic therapeutic leads.

## Results

### Catalytic domain of HaAEP1 purified at pH 4

AEPs are synthesized as inactive precursors that have been shown to undergo irreversible auto-activation into their mature form on exposure to a low pH environment that resembles the acidic pH in the vesicles/vacuole where these proteins are active in vivo (*Dall and Brandstetter, 2013*; *Shafee et al., 2015*). In order to obtain an active form of a plant AEP, a ~51 kDa pro-HaAEP1 (residues 28–491) lacking an endoplasmic reticulum signal sequence and including a N-terminal His-tag was expressed in *Escherichia coli* and purified by nickel affinity chromatography before being activated at pH 4.0 overnight. The activated form of HaAEP1 was then further purified by size exclusion chromatography, enabling separation of the core domain from the 'cap' domain (*Zhao et al., 2014*), and crystal trials were undertaken (*Figure 2—figure supplement 1*). SDS-PAGE analysis of the ~38 kDa core domain peak revealed the disassociation of the core domain from the cap domain but also showed several bands of HaAEP1, suggesting the presence of several cleavage sites at the termini of the core domain, as seen previously for several AEPs (*Hara-Nishimura et al., 1998*; *Nguyen et al., 2014*; *Zhao et al., 2014*; *Harris et al., 2015*). Indeed sequence comparison reveals the conservation of several of these predicted cleavage sites but notably lacks a previously described C-terminal di-Asp motif (*Hiraiwa et al., 1999*) (*Figure 2—figure supplement 2*). This led us to hypothesize that Asp52, Asn57, Asn338, Asp356, and Asp358 might represent the dominant auto-catalytic cleavage sites in HaAEP1.

### HaAEP1 structure reveals subtle differences to dictate substrate specificity

Crystallization trials of the ~38 kDa activated HaAEP1 yielded diffraction quality crystals that diffracted to a resolution of 1.8 Å (*Table 1*). The crystal structure was solved by molecular replacement yielding a single monomer in the asymmetric unit, and revealed an active monomeric HaAEP1 (residues 58–338 with weak electron density for Asn338) that forms a canonical C13 caspase structure, with a central six-stranded β-sheet region confined by five α-helices (*Hara-Nishimura et al., 1993*; *Yamada et al., 2005*) (*Figure 2A*, *Figure 2—figure supplement 3*). The structure of HaAEP1 lacks the C-terminal cap domain and N-terminal His-Tag, displaying dimensions of approximately 44 Å x 42 Å x 39 Å. Sequence analysis of this core domain suggests that the aforementioned pro-domains are likely to have been auto-catalytically processed during maturation as the previously predicted Asn cleavage sites precede and follow the defined active structure.

HaAEP1 displays structural similarity to hAEP and a recently published structure of inactive OaAEP1 (PDB ID: 4N6O and 5H0I) with an r.m.s.d. value of 1.0 and 0.7 Å over 262 and 267 carbon alpha residues, respectively (*Holm and Rosenström, 2010*; *Yang et al., 2017*) (*Figure 2B*). Due to such topological conservation it is expected that subtle differences around the substrate binding pocket will define substrate specificity and catalytic efficiency. Indeed, comparison of these three structures reveals HaAEP1 exhibits a unique flexible extension, reflected by weak electron density, in the α5-β6 loop and differences between the residues that are local to the catalytic His and Cys and those that delineate the S3-S5 pockets (following the protease nomenclature defined by Schechter and Berger where the cleavage site residue is termed P1 and residues prior to and following the cleavage site are labeled P5-P2 and P1′-P2′, respectively, and where the corresponding binding sites on the protease are described as S5-S2′) (*Schechter and Berger, 1967*). Specifically, differences in the substrate pocket include residues YGT 249–251 in HaAEP1 (hAEP: YAC 217–219, OaAEP1: WCY 246–248), a bulky Trp232 in hAEP versus Leu271 in HaAEP1 and Leu268 in OaAEP1, and the presence of an additional proline prior to the $β_{IV}$-$β_V$ polyproline loop which orients E257 away from the S4 region in HaAEP1 (*Figure 2B*). In OaAEP1, the C247 residue at the entrance to the S4 pocket

**Table 1.** Crystallography data collection and refinement statistics.
Numbers in parenthesis refer to the highest resolution bin.

| Data collection | |
| --- | --- |
| Space group | P3$_1$ 2 1 |
| Unit cell dimensions | |
| a, b, c (Å) | 77.03, 77.03, 108.17 |
| α, β, γ (°) | 90.00, 90.00, 120.00 |
| Wavelength | 0.9537 |
| Resolution (Å) | 1.8 |
| R$_{merge}$ (%) | 6.0 (43.3) |
| I/σI | 14.7 (2.2) |
| Completeness (%) | 93.8 (65.1) |
| Redundancy | 4.1 (1.8) |
| CC $_{1/2}$ | 0.997 (0.727) |
| **Refinement** | |
| Resolution (Å) | 66.71–1.80 |
| No. reflections | 31205 |
| R$_{work}$/R$_{free}$ | 15.15/18.88 |
| No. Atoms | 2415 |
| Protein | 2168 |
| Water | 229 |
| Ligand | 18 |
| Wilson B (Å$^2$) | 15.7 |
| **Average refined B-factor (Å$^2$)** | |
| Protein only (Å$^2$) | 22.2 |
| Water (Å$^2$) | 35.0 |
| Ligand (Å$^2$) | 41.1 |
| **r.m.s. deviations:** | |
| Bond lengths (Å) | 0.018 |
| Bond angles (°) | 1.84 |
| **Ramachandran analysis** | |
| Favored (%) | 98.84 |
| Allowed (%) | 1.16 |
| Outliers (%) | 0 |

DOI: https://doi.org/10.7554/eLife.32955.008

was recently proposed to function as a 'gate keeper for ligation' with large bulky side chains inhibiting ligation (*Yang et al., 2017*). Differences local to the active site include residues P181, Q245, N247 in the β$_I$ sheet and β5-β$_{IV}$ loop (OaAEP1: A178, T242, S244, hAEP: T151, R213, S215) and G185, E189, H191 in the β$_{II}$-β$_{III}$ region (OaAEP1: G182, K186, Y188, hAEP: V155, N158, D160) (*Figure 2B*).

A 6-residue insertion in the α5-β6 loop results in the disruption of a potential N-linked glycosylation site that was hypothesized to affect substrate binding upstream to P5 in hAEP (*Dall and Brandstetter, 2016*). Interestingly, none of the four conserved potential N-linked glycosylation sites in hAEP and mouse AEP (mAEP) are found in HaAEP1 (*Figure 2—figure supplement 2*). Given that these glycosylation sites have been predicted to protect AEP from non-specific protease activation it is intriguing to find that the only two potential N-linked glycosylation sites in HaAEP1 (N138 and N143) are located on the opposite side of the protein to the activation peptide in mammalian AEPs,

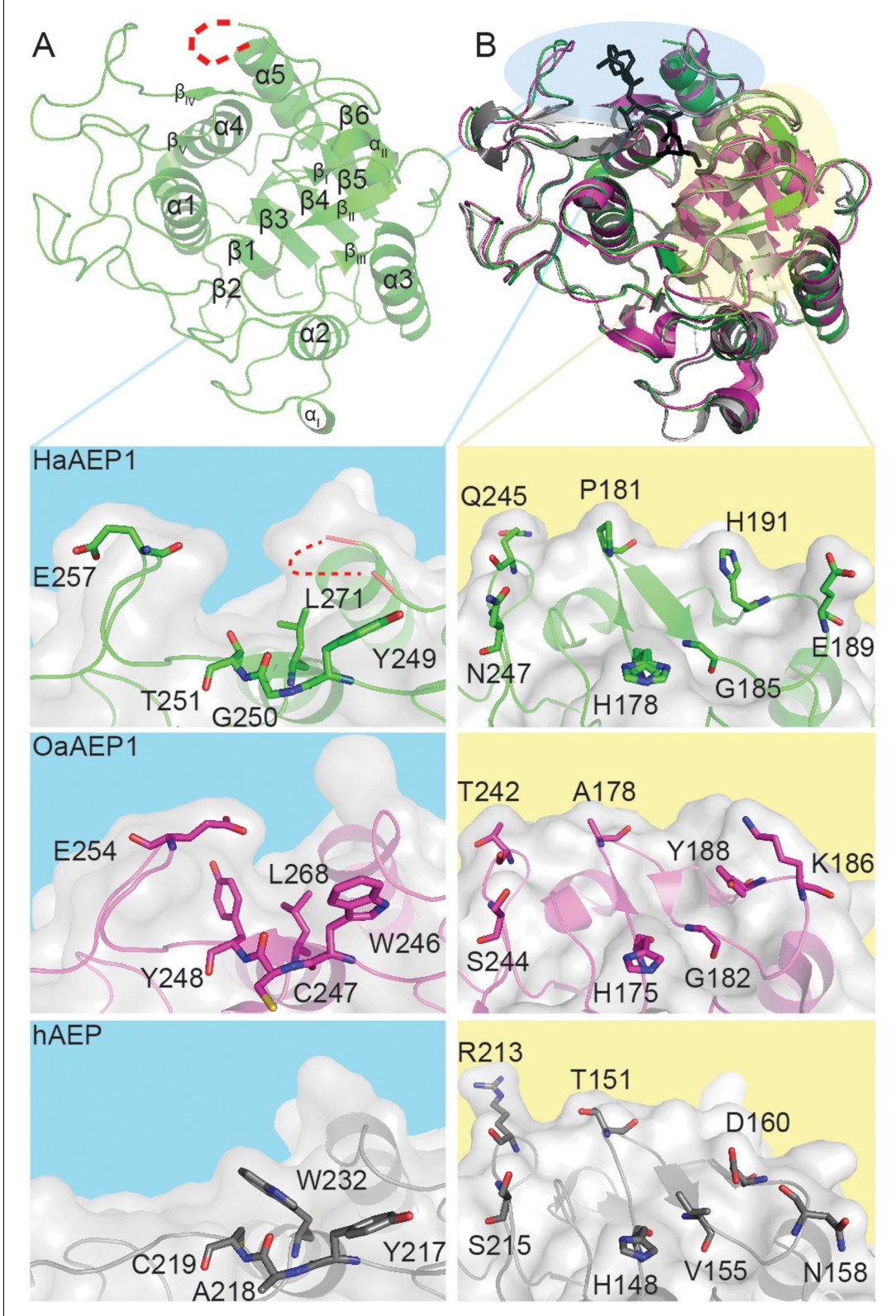

**Figure 2.** Architecture of HaAEP1 versus other AEP structures. (**A**) Cartoon representation of the overall topology of HaAEP1 with major α-helices and β-sheets annotated. Residues 304–309 that exhibited weak electron density and are missing from the model are shown as red dotted loop. (**B**) Comparison of HaAEP1 core domain (green) with OaAEP1 (magenta) and hAEP (gray) with bound chloromethylketone inhibitor (black) illustrating high overall structural similarity. Expanded surface and cartoon representations of highlighted regions of β$_{IV}$-β$_V$ substrate binding region (blue backgrounds)

*Figure 2 continued on next page*

*Figure 2 continued*

and catalytic region (β$_I$ sheet, β5-β$_{IV}$ loop and β$_I$-β$_{III}$ region) orientated over the catalytic His residue (yellow backgrounds) are shown below illustrating the residue differences that could alter substrate specificity (shown in stick format). Also see *Figure 2—figure supplements 1–3*.

DOI: https://doi.org/10.7554/eLife.32955.004

The following figure supplements are available for figure 2:

**Figure supplement 1.** HaAEP1 auto-catalytically activates upon pH shift to pH 4.

DOI: https://doi.org/10.7554/eLife.32955.005

**Figure supplement 2.** Sequence alignment of C13 family of cysteine proteases.

DOI: https://doi.org/10.7554/eLife.32955.006

**Figure supplement 3.** Topology diagram of active HaAEP1.

DOI: https://doi.org/10.7554/eLife.32955.007

at the beginning and center of the α2 helix, respectively (*Dall and Brandstetter, 2016*). Moreover, only N138 is prominently surface exposed, suggesting that N-linked glycosylation in HaAEP1 is not utilized for mitigation against non-specific premature activation.

## HaAEP1 displays a tetrahedral intermediate in the active site

The 1.8 Å resolution of our HaAEP1 structure allowed us to observe a succinimide moiety (Snn) below the catalytic His in a location identical to human AEP (*Figure 3A*) that was hypothesized to play a role in peptide ligation (*Dall et al., 2015*). We distinguished dual conformations of the

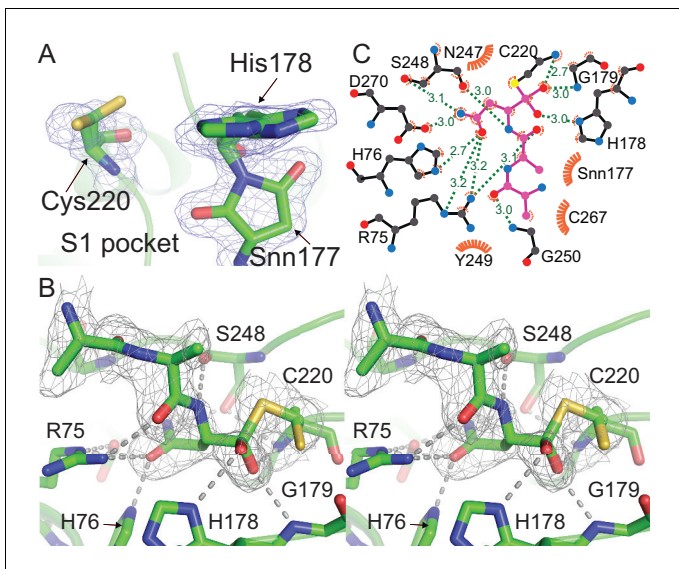

**Figure 3.** Outstanding features of the HaAEP1 active site. (A) Catalytic residues with dual conformations illustrated in the HaAEP1 active site with simulated annealing omit electron density maps (2 $F_{obs}$ - $F_{calc}$) contoured at 1σ level. (B) Cross-eyed stereo view of polder OMIT map ($F_{obs}$ - $F_{calc}$) calculated in the absence of the shown overlaid AAN tetrahedral intermediate, contoured at 3σ level. (C) Schematic representation of interactions between AAN tetrahedral intermediate (purple sticks) and labeled active site residues generated using LigPlot$^+$ (*Laskowski and Swindells, 2011*). Residues forming hydrogen bonds (green) are illustrated with black sticks with distances shown (Å). Residues and atoms that provide hydrophobic interactions are highlighted with orange eyelash symbols. Also see *Figure 3—figure supplement 1*.

DOI: https://doi.org/10.7554/eLife.32955.009

The following figure supplements are available for figure 3:

**Figure supplement 1.** HaAEP1 active site geometry.

DOI: https://doi.org/10.7554/eLife.32955.010

**Figure supplement 2.** Polder OMIT maps define the presence of a tetrahedral intermediate in the HaAEP1 active site.

DOI: https://doi.org/10.7554/eLife.32955.011

catalytic His178 and Cys220 residues which we hypothesize represent conformational changes that occur during catalysis and correspond to substrate free and reaction intermediate states (*Figure 3A*). In the intermediate state the catalytic Cys Sδ is oriented ~95° towards the Nδ of the catalytic His imidazole ring which is orientated ~3.8 Å closer to this residue in the corresponding intermediate state. Moreover, flexibility of His in a relatively open pocket free of steric hindrance (*Figure 3—figure supplement 1*) suggests an additional role for conformational shifts of the His in catalysis as seen with a range of proteases (*McLuskey et al., 2012*; *Clark, 2016*; *Chekan et al., 2017*). In the proposed alternate resting state the catalytic Cys Sδ is oriented towards the backbone amine of the highly conserved Gly179, reducing the distance between them from 5.4 Å to 3.6 Å. In this orientation the backbone amine might function in stabilizing proton abstraction from the Cys thiol.

Close examination of the electron density in the active site of HaAEP1 suggested a small peptide chain is intermittently bound to the intermediate conformation of Cys220 (*Figure 3B*). Given AEPs specificity for Asx and the nature of the electron density, we built and refined a tetrahedral complex of a three-residue peptide ligand (AAN) bound to the active site Cys of HaAEP1 with partial occupancy (*Figure 3B*). Residues Ala-Ala were modeled upstream of the P1 Asn due to the weak electron density away from the peptide backbone at these residues. The AAN peptide ligand allowed characterization of interactions likely to exist between HaAEP1 and P1 Asn and main chain of residues P2-P3 during substrate recognition. Due to the high sequence conservation of the HaAEP1 active site with mammalian AEPs and conserved substrate orientation, many of the interactions with the substrate match those observed for inhibitors of hAEPs (*Dall et al., 2015*) (*Figure 3C*). The presence of this unexpected substrate in the HaAEP1 active site is likely to be a product of auto-activation. The continuous electron density between Cys220 and the P1 Asn supports the interpretation of the formation of a tetrahedral intermediate that is stabilized by the presence of an oxyanion hole formed by His178, Gly179 and the backbone amine of Cys220 that is more congruous to the electron density than an acyl intermediate or free peptide (*Figure 3B*, *Figure 3—figure supplement 2*). Moreover, the observed short distance between the tetrahedral intermediate carbon atom and catalytic cysteine sulfur atom is incompatible with the absence of a covalent interaction. Main chain amino groups of Gly residues have previously been proposed to function in the creation of an oxyanion hole stabilizing the formation of a tetrahedral intermediate with the substrate in a wide range of cysteine proteases (*Dall and Brandstetter, 2016*). Furthermore, previous observations of tetrahedral intermediates and enzyme-product complexes with serine proteases have been shown to exhibit a pH-dependent equilibrium (*Wilmouth et al., 2001*; *Radisky et al., 2006*; *Lee and James, 2008*). Similarly, the trapping of this intermediate state may have been fortuitously facilitated by the activation of HaAEP1 at pH 4, followed by crystallization at pH 7. Although alternative conformations of the active site Cys in hAEP have been described previously, this tetrahedral intermediate state has not been described before (*Dall et al., 2015*).

## Modulation of pH enables HaAEP1 to perform macrocyclization

The similarity of the HaAEP1 active site to that of OaAEP1 and the revelation that it contained a reactive succinimide prompted us to test an enzyme preparation similarly taken to pH 4.0 against the modified SFTI-1 precursor substrate SFTI(D14N)-GLDN substrate. HaAEP1 had previously been unable to create a macrocyclic product from SFTI(D14N)-GLDN, but had been shown to efficiently cleave it at a rate, $k_{cat}/K_m$ value of 610 $M^{-1}$ $S^{-1}$, similar to rates published for other AEPs (*Bernath-Levin et al., 2015*). To our surprise, our new preparations of HaAEP1 taken to pH 4.0 produced cyclic SFTI(D14N) when the reactions were conducted at pH 6.5 (*Figure 4B* WT, *Figure 4—figure supplement 1A* WT). Previously, HaAEP1 had been purified at pH 8, activated at pH five and then used in reactions at pH 5. The HaAEP1 preparations that were able to macrocyclize were similarly purified at pH 8, but activated at pH four then returned to pH 6.5. Activation at lower pH has been shown to be more effective at removing the cap domain, which in mammalian AEPs had a propensity to re-ligate (*Zhao et al., 2014*). Higher pH could also favor macrocyclization by facilitating the deprotonation of the Gly N-terminus at the active site, priming it to attack its C-terminus, which is held in the thioacyl intermediate at the active site.

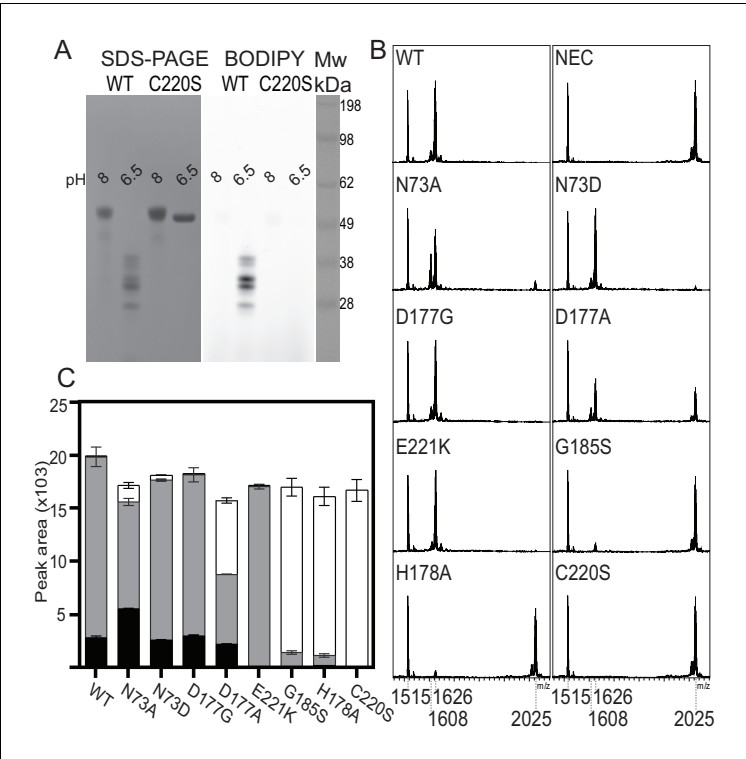

**Figure 4.** Mutagenesis studies of HaAEP1. (**A**) Activity based probe analysis of HaAEP1 WT and C220S mutant illustrates that WT is active at pH 6.5 after activation at pH 4 whereas C220S mutant remains inactive. In-gel fluorescence of activity based probe (right) and post-fluorescence Coomassie stain of SDS-PAGE gel (left). (**B**) MALDI-TOF spectra of SFTI(D14N)-GLDN processing by WT and a range of HaAEP1 mutants directed at altering the ability of HaAEP1 to cleave and macrocyclize the substrate SFTI(D14N)-GLDN. NEC – no enzyme control. (**C**) Quantitation of peak areas from **B**. Peak areas of mass 1608 - cyclic SFTI(D14N), mass 1626 - acyclic-SFTI(D14N) and mass 2025 - unprocessed seleno-Cys modified SFTI(D14N)-GLDN substrate, were normalized for ionization efficiency using an internal standard mass 1515 - native SFTI-1. Black - cyclic SFTI-1(D14N), gray - acyclic-SFTI (D14N) and white - unprocessed seleno-Cys modified SFTI(D14N)-GLDN substrate. Peak areas with acyclic and cyclic forms have previously been shown to exhibit similar ionization efficiencies (*Bernath-Levin et al., 2015*). Error bars illustrate standard deviation n = 3 (D177A n = 2) technical replicates. Also see *Figure 4—figure supplements 1–4*.

DOI: https://doi.org/10.7554/eLife.32955.012

The following figure supplements are available for figure 4:

**Figure supplement 1.** Analysis of WT HaAEP1 and mutants activity.
DOI: https://doi.org/10.7554/eLife.32955.013
**Figure supplement 2.** Model of SFTI-1 processing from PawS1.
DOI: https://doi.org/10.7554/eLife.32955.014
**Figure supplement 3.** Circular dichroism analysis of HaAEP1 secondary structure.
DOI: https://doi.org/10.7554/eLife.32955.015
**Figure supplement 4.** Re-ligation of cap domain at pH 8 favored after activation at pH 6.5.
DOI: https://doi.org/10.7554/eLife.32955.016

## Residues local to the catalytic dyad influence product formation

To investigate the mechanism of macrocyclization by HaAEP1 we identified several residues for site directed mutagenesis. Firstly, we hoped to clarify the roles of Cys220 and Snn177 in macrocyclization through Ser and Ala mutations, respectively. Furthermore, we also mutated Snn177 to Gly, as the C13 protease GPI8 has also been proposed to carry out an intramolecular transpeptidation reaction yet displays a Gly residue at a location equivalent to Snn177 (*Zacks and Garg, 2006*) (*Figure 2—figure supplement 2*). Secondly, we hoped to alter the ability of HaAEP1 to macrocyclize its native substrate SFTI-1 by altering the residues that fine tune this catalysis. By modeling the binding of an

NMR structure of PawS1 (*Franke et al., 2017*) and the N-terminally cleaved SFTI-1 precursor (SFTI-GLDN) to HaAEP1 we were able to hypothesize the location of a hydrophobic S2' binding region encompassing strand $\beta_{II}$ with G185 located at the center of this region (*Figure 4—figure supplement 2*). In addition, previous studies have also implicated that Asn, Glu and Asp residues proximal to the catalytic Cys and His function in catalysis (*Dall and Brandstetter, 2013*; *Zhao et al., 2014*). Inspection of the HaAEP1 structure revealed that several of these residues (including Asn73 and Glu221) are conserved in HaAEP1. Glu221 is oriented away from Cys220, in a direction similar to that seen in hAEP bound to human cystatin E, it might assume alternate conformations due to its solvent exposure and high B-factor.

The mutant HaAEP1 proteins were expressed in *E.coli* and analyzed by circular dichroism (*Figure 4—figure supplement 3*), with wild-type (WT) HaAEP1 displaying a melting temperature of ~52°C similar to previous reports for hAEP (*Dall and Brandstetter, 2013*), and mutants displaying similar spectra to WT.

Incubation of WT HaAEP1 with a fluorophore labeled (BODIPY) activity-based probe (*Lu et al., 2015*) illustrated its heterogeneity in size following activation at pH 4 and incubation with substrate at pH 6.5, which has previously been observed for AEP proteins both in vivo and in vitro and speculated to be the result of processing of non-glycosylated forms (*Zhao et al., 2014*) (*Figure 4a*). This probe also revealed a C220S mutation in HaAEP1 to result in the enzyme becoming incapable of pH-dependent activation (*Figure 4a*). Further analysis of AEP mutants activity using the BODIPY probe illustrates the substantial heterogeneity in size between active WT and N73A, N73D, D177G, D177A or E221K mutant AEPs, likely due to autocatalytic processing, yet indicates that they remain active (*Figure 4—figure supplement 1D*).

The use of a seleno-modified synthetic SFTI(D14N)-GLDN substrate, which we have previously shown to be processed by HaAEP1 and CeAEP1(*Bernath-Levin et al., 2015*), allowed for a comparison of activity profiles of HaAEP1 mutants through the quantification of distinctive isotopic cyclic, acyclic and unprocessed peak areas by MALDI-MS (*Figure 4B–C*, *Figure 4—figure supplement 1A*). This comparison reveals subtle changes between mutants in the ratio of cyclic, acyclic and unprocessed peptides and confirmed that the HaAEP1-C220S mutant is unable to cleave as evidenced by the lack of a peak at mass 1608 or 1626 (*Figure 4*, *Figure 4—figure supplement 1*). As expected, mutation of the second residue of the catalytic dyad (H178A) also results in a drastic reduction in activity, based on SFTI(D14N)-GLDN processing and activity based BODIPY probe results, confirming the significance of C220 and H178 in AEP activity (*Figure 4*, *Figure 4—figure supplement 1*). Interestingly, the H178A mutation does not abolish HaAEP1 activity, as previously seen with mAEP, and suggests a third residue could facilitate proton transfer at the active site (*Zhao et al., 2014*). Similar results to the H178A mutation were also observed for G185S that was directed at altering the hydrophobic S2' binding region. The effect of the G185S mutation suggests G185 has a role in substrate recognition and could sterically alter the conformation of H178 and Snn177 (*Figure 4*, *Figure 3—figure supplement 1*). The mutation E221K, which has previously been shown to increase endopeptidase activity in hAEP, resulted in a loss of cyclic product as shown by an absence of a peak of mass 1608 (*Figure 4B*, *Figure 4—figure supplement 1A*). Mutation of N73A leads to a higher ratio of cyclic to acyclic product as shown by an increased peak area relative to WT of mass 1608 and a reduced acyclic product peak of mass 1626. N73D and D177G mutants appear to process SFTI(D14N)-GLDN in a manner similar to WT. Whereas the large fraction of unprocessed SFTI(D14N)-GLDN, mass 2025, after incubation with D177A illustrates a reduction in processing efficiency, as previously noted for the mAEP D149A mutant (*Zhao et al., 2014*).

The effect of the N73A, E221K and D177G mutations on AEP catalyzed ligation was probed further by investigating each mutant's ability to revert to its inactive form via re-ligation of its cap domain upon shift to neutral pH. As described for mAEP and equivalent mutants (*Zhao et al., 2014*), incubation of HaAEP1 at pH 4 led to an irreversible dissociation of its cap domain (*Figure 4—figure supplement 4*). However, following activation at pH 5.5 and 6.5, upon shifting to pH 8 the WT, N73A, E221K and D177G mutants were able to re-ligate the cap onto the core domain resulting in the formation of the inactive pro-enzyme with a Mw ~52 kDa as evidenced by SDS-PAGE and activity based probes (*Figure 4—figure supplement 4*).

These results confirm the importance of the catalytic dyad in AEP function and show that HaAEP1, like mAEP and the closely related C13 family member GPI8, is able to perform its ligation

reaction in the absence of a Snn residue. Moreover, subtle mutations affecting the stability of the catalytic dyad might favor either hydrolysis or macrocyclization.

## Active HaAEP1 exhibits an open surface amenable to macrocyclization

To discern the structural determinants that favored AEPs to be recruited independently by evolution multiple times for macrocyclization, we compared the structure of HaAEP1 and its predicted binding mode with the crystal structures of closely related cysteine proteases: sortase A, papain and meta-caspase MCA2 (*Suree et al., 2009*; *Chu et al., 2011*; *McLuskey et al., 2012*) (*Figure 5*).

Sortase A is a *Staphylococcus aureus* cysteine protease which catalyzes a similar transpeptidation reaction to AEP, ligating proteins bearing a LPXTG motif to peptidoglycan precursors in the bacterial cell wall (*Mazmanian et al., 1999*). NMR studies have shown that the resolution of the transpeptidation thioacyl intermediate reaction occurs through nucleophilic attack of a lipid terminal amine in

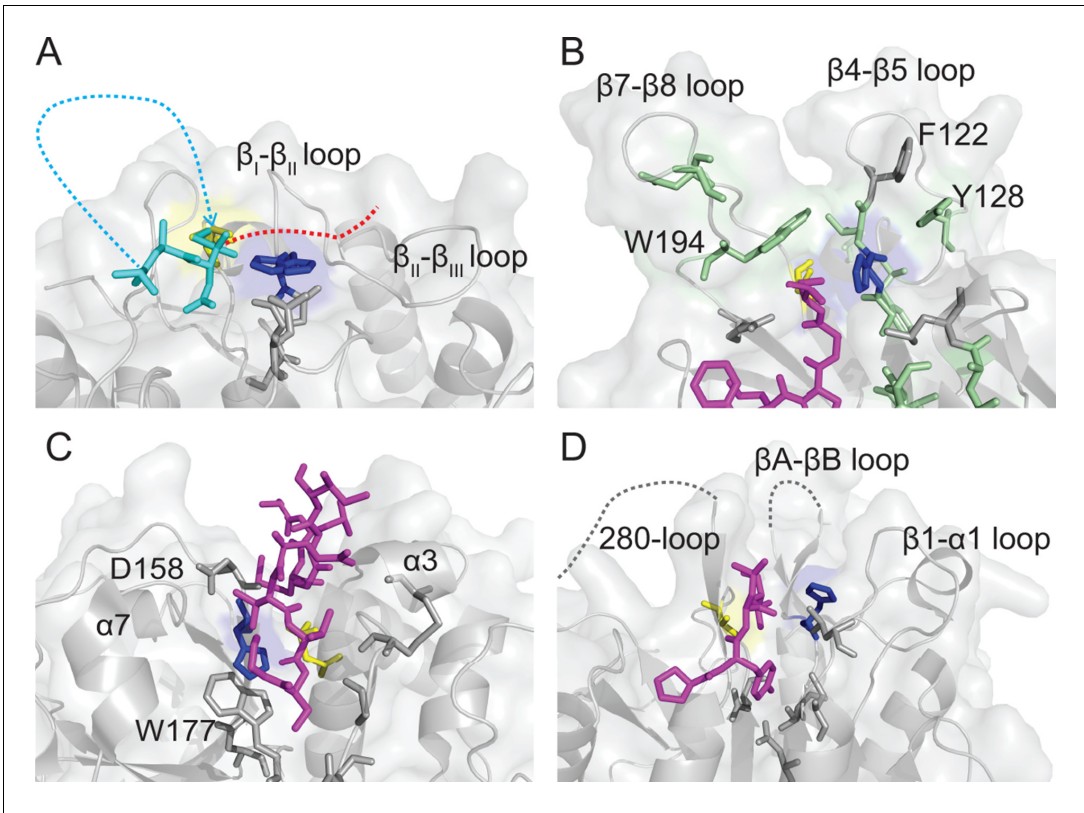

**Figure 5.** Comparison of Cys protease active site topologies. Substrates are oriented towards the catalytic dyad of Cys and His, highlighted in yellow and blue sticks, respectively. Regions and residues likely to impart steric hindrance on the proposed mode of macrocyclization by AEPs in other Cys proteases are labeled. (**A**) Proposed binding mode for SFTI-1 macrocyclization (cyan sticks and dots) based on the position of the tetrahedral intermediate in the HaAEP1 structure with the N-terminus attacking the intermediate and the GLDN-tail (red-dots) oriented over the catalytic His towards the $\beta_{II}$-$\beta_{III}$ region. (**B**) Structure of sortase A (PDB 2KID) covalently bound to an analog of the sorting signal (purple sticks). Residues implicated in transpeptidation are highlighted with green sticks, in a region analogous to the $\beta5$-$\beta_{IV}$ loop and $\beta_I$-$\beta_{II}$ loop of HaAEP1. (**C**) Structure of papain (PDB 3IMA) bound to residues 2–7 and 49–53 of tarocystatin (purple sticks) illustrating restricted access to the catalytic dyad. (**D**) Structure of the metacaspase MCA2$_{C213A}$ (PDB 4AFV) with residues 30–33 of the N-terminal domain (purple sticks) bound in the predicted direction for substrate binding. Flexible regions of low electron density are shown with gray dots. Also see *Figure 5—figure supplement 1*.
DOI: https://doi.org/10.7554/eLife.32955.017

The following figure supplement is available for figure 5:

**Figure supplement 1.** Comparison of crystal structures of AEPs and a model of an efficient macrocyclizing AEP.
DOI: https://doi.org/10.7554/eLife.32955.018

a steep valley between the β7-β8 and β4-β5 loops (*Suree et al., 2009*). In comparison to HaAEP1, the sortase A β7-β8 loop protrudes much further from the active site and the aromatic residues F122, Y128, W194 in these loops orient over the catalytic Cys (*Figure 5B*). Together these loops, despite reported flexibility, would likely impart considerable steric hindrance for macrocyclization by inhibiting both the resolution of the intermediate by the N-terminal amine group and binding at the S2′ substrate 'tail'.

Papain from the melon tree *Carica papaya* is the archetypal plant C1 family cysteine protease and has been found to bind to a cystatin homolog, tarocystatin, in a manner analogous to that of hAEP binding cystatin (*Otto and Schirmeister, 1997*; *Chu et al., 2011*; *Dall et al., 2015*). However, inspection of the papain active site reveals a topology that is not conducive for macrocyclization with the catalytic triad buried deep within the protein and steric hindrance for peptide N-terminal attack likely to be imparted by W177, D158 and the extended alpha helical loops of the α3 and α7 regions (*Figure 5C*).

Metacaspases are expressed in plant, fungi and protozoa and display a C14 caspase domain that is structurally homologous to human caspases (*Tsiatsiani et al., 2011*). Metacaspases reside within the same CD clan as AEP, but exhibit a strict specificity for a cleavage following Arg or Lys (*Vercammen et al., 2004*). Currently no crystal structure is available for a plant metacaspase, however the crystal structure of the protozoan metacaspase MCA2 reveals an architecture that like papain would likely be unfavorable for macrocyclization due to steric hindrance around the active site from several prominent loops; including the β1-α1, βA-βB and 280-loop (*McLuskey et al., 2012*) (*Figure 5D*).

The crystal structure of active HaAEP1 suggests that the combination of a relatively open reaction interface with space around the active site allowing for catalytic residue flexibility has resulted in the convergence upon AEPs for macrocyclization ahead of the other 30 families of cysteine proteases in plants (*Rawlings et al., 2016*).

## AEPs have an intrinsic ability to perform peptide macrocyclization

Given the sequence similarity between AEPs and the conservation of residues involved in catalysis we hypothesized that the ability to macrocyclize peptides might be inherent to AEPs. To test this hypothesis we recombinantly expressed two AEPs from species which are currently not thought to contain cyclic peptides of any kind; *Arabidopsis thaliana* (AtAEP2) and *Ricinus communis* (RcAEP1), respectively. These AEPs were purified and activated as described for HaAEP1 (pH 4) and then incubated with non-native substrates; SFTI-GLDN and SFTI(D14N)-GLDN, at a pH that favors ligation (pH 6.5). Under these conditions RcAEP1 was able to macrocyclize both SFTI-GLDN and SFTI (D14N)-GLDN substrates whereas AtAEP2 was able to macrocyclize only SFTI(D14N)-GLDN (*Figure 6*). These findings further support our hypothesis that the structural features of AEPs described above have allowed for the convergence upon AEP for peptide macrocyclization reactions.

## Discussion

Herein, we have described the structure of an active plant AEP containing a peptide ligand with partial occupancy bound to the active site catalytic Cys as a tetrahedral intermediate, illustrating conformational flexibility in the AEP catalytic dyad. This structure has enabled us to predict a model for SFTI-1 macrocyclization by HaAEP1 where the GLDN tail of the SFTI-1 precursor orients in a manner analogous to cystatin binding to hAEP, over the catalytic His to a hydrophobic patch on the $β_{II}$ region, and where the N-terminus attack occurs between the diminutive $β_I$ sheet and β5-$β_{IV}$ loop in a manner analogous to the attack of thioacyl intermediates by peptidoglycan precursors in sortase-catalyzed transpeptidation reactions (*Suree et al., 2009*; *Clancy et al., 2010*). This model complements the requirement for a small amino acid followed by a hydrophobic residue at P1′ and P2′ due to their orientation over the catalytic His and towards the hydrophobic $β_{II}$-$β_{III}$ region, respectively.

For several Cys proteases, including AEPs, a minor role for a third Asx/Arg residue in catalysis has previously described (*Vernet et al., 1995*; *Dodson and Wlodawer, 1998*; *Dall and Brandstetter, 2016*; *Clancy et al., 2010*; *Buller and Townsend, 2013*; *Dall and Brandstetter, 2013*). This third residue has been suggested to function in stabilization and orientation of the His imidazole ring. Interestingly, functional analysis of HaAEP1 catalytic triad residue N73 revealed that an N73A mutation resulted in the production of a higher ratio of cyclic SFTI-1. The observation of potential

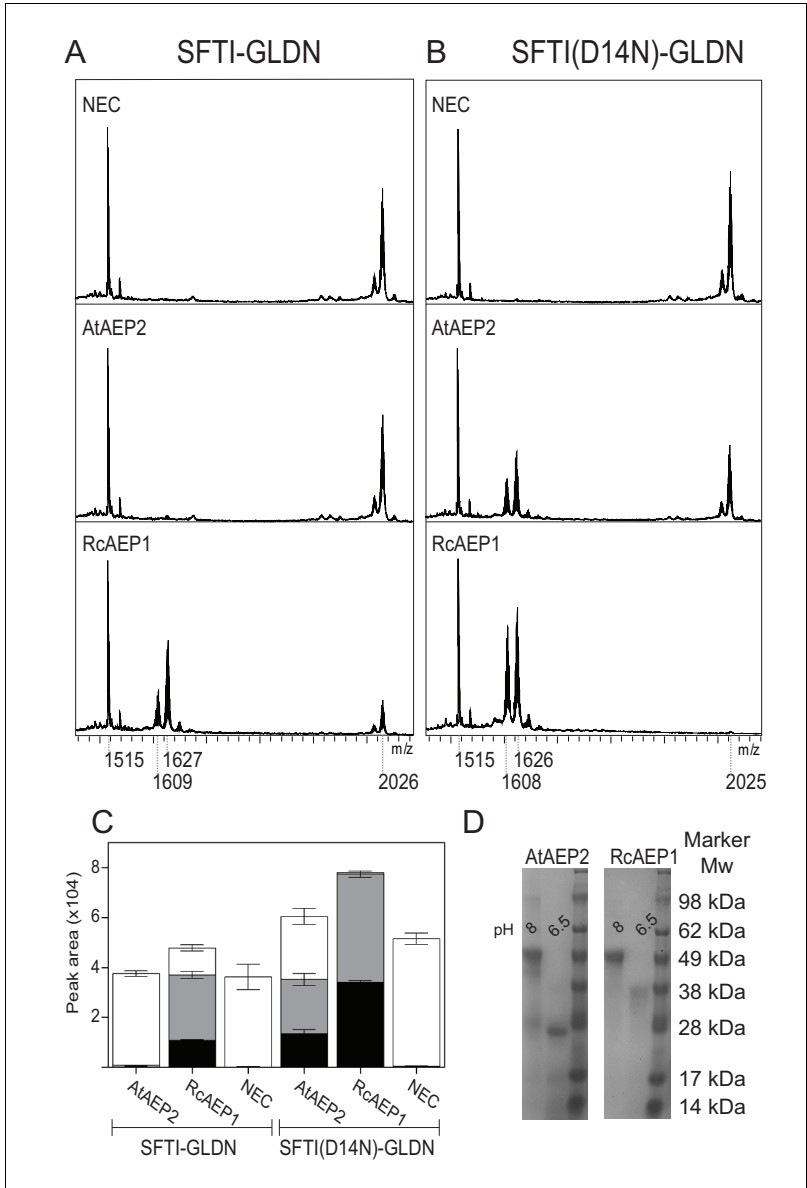

**Figure 6.** The ability to macrocyclize is inherent to AEPs. MALDI-TOF spectra of SFTI-GLDN (**A**) and SFTI(D14N)-GLDN (**B**) with AEPs from species thought not to contain cyclic peptides, namely *Arabidopsis thaliana* (AtAEP2) and *Ricinus communis* (RcAEP1). (**C**) Quantitation of peak areas from seleno-Cys modified substrates in A) (mass 1609 - cyclic SFTI-1, mass 1627 - acyclic-SFTI and mass 2026 - unprocessed SFTI-GLDN substrate) and B) (mass 1608 - cyclic SFTI(D14N), mass 1626 - acyclic-SFTI(D14N) and mass 2025 - unprocessed seleno-Cys modified SFTI (D14N)-GLDN substrate) normalized to mass 1515 - native SFTI-1. Black – cyclic peptide, gray – acyclic peptide and white – unprocessed substrate. Error bars illustrate standard deviation n = 3 technical replicates. (**D**) SDS-PAGE analysis of inactive (pH 8) and active (pH 6.5) AtAEP2 and RcAEP1 proteins.
DOI: https://doi.org/10.7554/eLife.32955.019

conformational shifts in the crystal structure of HaAEP1 during catalysis suggests that the loss of an orientating N73 side chain allows for further conformational flexibility of H178. This increased flexibility of His in a relatively open pocket that might accommodate a range of rotamers could reduce steric hindrance of an N-terminus entry towards the acyl intermediate and also facilitate deprotonation of the substrate N-terminus.

The requirement for space and flexibility between the catalytic dyad favors the convergence for peptide macrocyclization upon Cys proteases over Ser proteases due to the close proximity between

Ser-His residues in these proteases, an inherent requirement based upon the reduced nucleophilic properties of Ser (*Buller and Townsend, 2013*). Indeed, a recent structure of a macrocyclizing Ser protease, PCY1, suggested that a shift in the catalytic His away from Ser is required for macrocyclization, based on a comparison to their hydrolytic relatives, and suggested a role for the catalytic His in deprotonation of the attacking peptide N-terminus (*Chekan et al., 2017*). Furthermore, comparison with other Cys proteases suggests that AEPs have been converged upon for macrocyclization based on their relatively flat and open catalytic site. In contrast to a recent hypothesis for efficient macrocyclization based on a comparison of AEP structures focused on a region closer to the S4 pocket, here we suggest that differences in AEP catalytic efficiency and substrate specificity will be defined by subtle amino acid differences in the $\beta_{II}$-$\beta_{III}$ region, $\beta_I$ sheet and $\beta5$-$\beta_{IV}$ loop and the S1-S5 pocket (*Yang et al., 2017*). Moreover, this orientation of the tail away from the $\beta_I$ sheet and $\beta5$-$\beta_{IV}$ loop is hindered in other caspases which exhibit a relatively straight substrate channel (*Figure 5*) and thus prevent simultaneous attack of the thioacyl intermediate by the N-terminus loop upon scissile bond cleavage.

Following activation, AEP functionality has been illustrated to be dependent on a delicate balance between pH and stability, where endopeptidase activity is favored at a low pH (~pH 4) and ligase activity is favored at a higher pH (~pH 6) (*Dall et al., 2015*). A pH-dependent activity has been well documented for cysteine proteases and is a function of the formation of a thiolate and imidazolium ions on the catalytic dyad of Cys and His residues (*Dall and Brandstetter, 2016*; *Frankel et al., 2005*). Given the hypothesized role of His in deprotonating the attacking N-terminal peptide chain in peptide macrocyclization it would therefore be expected that at a low pH the catalytic His would be more readily protonated and hence this reaction might be less frequent resulting in more hydrolysis at low pH. This hypothesis is supported by the low level of endopeptidase activity observed with Ala mutations of the catalytic His in HaAEP1 and other proteases, illustrating that thiolates might form and that hydrolytic resolution of the thioacyl intermediate might occur in the absence of His (*Frankel et al., 2007*; *Zhao et al., 2014*). Previous reports have also suggested that a local Gly amide backbone might facilitate catalysis via a transfer of a proton to the leaving group of the tetrahedral intermediate from a hydrogen bonded water molecule; analysis of the HaAEP1 structure reveals Gly179 could potentially perform this role in the absence of His (*Brady et al., 1999*). Moreover, mutation of the acidic residue following the catalytic Cys to a basic residue in hAEP has previously been shown to enhance catalytic efficiency by decreasing the local pKa of the Cys residue (*Dall and Brandstetter, 2013*). Interestingly, we found an equivalent mutation in HaAEP1 to result in the loss of peptide macrocyclization indicating that E221 could also aid in deprotonation of the incoming N-terminus or that the residue's larger side chain and expected orientation in activating the catalytic Cys could impart steric hindrance upon the N-terminus attack of the thioacyl intermediate (*Figure 4*, *Figure 4—figure supplement 1*). However, given the finding that E221K mutation is unable to prevent re-ligation of the cap domain to the active core domain upon pH shift in HaAEP1, there is likely to be redundancy between the local residues in creating a nucleophilic amine group to complete a transpeptidation reaction.

The finding that AEPs from species that lack cyclic peptides may be coaxed into performing peptide macrocyclization of a linear peptide under favorable conditions significantly expands the potential use of AEPs for the production of cyclic peptides (*Figure 6*). Moreover, further investigation into the subtle nuances that define substrate specificity and catalytic activity between AEPs is warranted, with differences in the $\beta_{II}$-$\beta_{III}$ region, $\beta_I$ sheet, $\beta5$-$\beta_{IV}$ loop and around the S1-5 pocket including the variable $\alpha5$-$\beta6$ and $\beta_{IV}$-$\beta_V$ loops likely to be key (*James et al., 2017*). Indeed, in comparison to HaAEP1 and OaAEP1, the efficient peptide macrocyclizing AEP butelase one displays several different amino acids that could be responsible for this AEPs efficacy. These differences include shorter sidechains in the $\beta_I$ sheet and $\beta5$-$\beta_{IV}$ loops that may reduce steric hindrance on a peptides N-terminus during attack on a thioacyl intermediate (HaAEP1: P181, Q245, N247 OaAEP1: A178, T242, S244 Butelase-1: A168, G232, S234) and differences in the $\beta_{IV}$-$\beta_V$ region that could generate substrate specificity (*Figure 5—figure supplement 1*).

Active HaAEP1 exhibits a succinimide at the same position as hAEP which has previously been postulated to perform a Cys-independent ligation reaction through cyclic rearrangement with the P1 Asx side chain (*Dall et al., 2015*). Functional analysis reveals that HaAEP1 is able to perform peptide macrocyclization despite D177G/A mutations that cannot form a succinimide group (*Figure 4B–C*). Moreover, these mutants continued to perform re-ligation of the cap domain when activated at

pH >5.5 (*Figure 4—figure supplement 4*), a result that has previously been shown with mAEP (*Zhao et al., 2014*). In the absence of a critical requirement for succinimide formation in macrocyclization the question remains as to why this relatively unstable aspartimide appears stable in AEP crystal structures and is largely conserved in the C13 proteases. Succinimides have been shown to form more readily when adjacent to His residues as the His Nδ abstracts a proton from the His backbone NH allowing the deprotonated main chain amine to attack the Asx side chain (*Brennan and Clarke, 1995*; *Takahashi et al., 2016*). Once formed this succinimide could enhance the activity of the catalytic His by virtue of reducing stabilizing hydrogen bonding interactions with the carboxyl terminus and be involved in the proper positioning of the catalytic His. A further potential role for succinimides could present upon their hydrolysis through racemization and favored formation of iso-Asp (*Geiger and Clarke, 1987*; *Reissner and Aswad, 2003*). This iso-Asp could potentially represent a subtle mode of AEP regulation as the orientation of this side chain towards the S1 pocket is likely to disrupt substrate binding. Such iso-Asp residues have previously been proposed to regulate protein activity by a time-dependent molecular switch (*Geiger and Clarke, 1987*; *Ritz-Timme and Collins, 2002*).

In the absence of caspases, plants have evolved a wide range of cysteine proteases to ensure functional redundancy in a myriad of highly regulated programmed cell death pathways in response to environmental and developmental cues (*Fagundes et al., 2015*). Furthermore, plants have developed a variety of strategies to control the destructive prowess of these proteases including the expression of proteases as inactive zymogens with cofactor dependency, compartmentalization and the production of protease inhibitors (*Martínez et al., 2012*). Of these cysteine proteases the AEPs have recently attracted considerable interest due to their ability to carry out peptide macrocyclization and their potential application in the synthesis of pharmacoactive cyclic peptides. Herein, the structural and functional analysis of HaAEP1 has revealed residues that are able to favor the production of cyclic or acyclic products from SFTI(D14N)-GLDN. Moreover, we have modeled a binding mode for productive macrocyclization of the HaAEP1 natural ligand SFTI-1, based on a comparison with related cysteine proteases that is likely to be conserved between AEPs, where substrate specificity is defined by the amino acids around the binding site for respective AEPs. Furthermore, we have shown that AEPs from diverse species lacking cyclic peptides are able to perform macrocyclization under favorable pH conditions. These findings provide the foundation for further optimization of AEPs, potentially widening the array of peptide substrates that could be cyclized by AEPs.

## Materials and methods

### Protein expression and purification

DNA sequence encoding residues 28–491 of HaAEP1 (accession code: KJ147147), *Ricinus communis* AEP (RcAEP1) residues 58–497 (accession code: D17401) and *Arabidopsis thaliana* AEP (AtAEP2) residues 47–486 (accession code: Q39044) were cloned into a pQE30 (QIAGEN, Hilden, Germany) expression vector with an N-terminal six-histidine tag and expressed in the SHuffle strain *E. coli* (New England Biolabs) transformed with pREP4 (QIAGEN). Briefly, cultures were grown at 30°C to an $OD_{600}$ of 0.8–1.0 in Luria Broth medium containing 100 µg/mL ampicillin and 35 µg/mL kanamycin with expression induced at 16°C with 0.1 mM isopropyl β-D-1-thiogalactopyranoside then cells cultured overnight. Cell pellets were collected by centrifugation and lysed by ultrasonication in 50 mM Tris (pH 8.0), 100 mM sodium chloride, 0.1% Triton X-100. The soluble fraction was then harvested by centrifugation and the supernatant was incubated (batch wise) with Ni-NTA resin overnight at 4°C. The resin was then washed with 20 mL of 50 mM Tris (pH 8.0) and 20 mL of 50 mM Tris (pH 8.0) 20 mM imidazole and protein was eluted stepwise with 20 mL of 50 mM Tris (pH 8.0) 300 mM imidazole. For purification for crystal screens, six-histidine tagged inactive protein was purified, using an ÄKTA FPLC platform, by anion-exchange chromatography (HiTrap Q HP 5 mL) with a gradient of 0 to 500 mM sodium chloride over 60 min and then either activated by dialysis in 100 mM citric acid - sodium citrate buffer (pH 4.0) containing 50 mM sodium chloride and 5 mM DTT, overnight at 16°C, or directly concentrated and further purified by size-exclusion chromatography (HiLoad 16/600 Superdex 200) in 50 mM Tris (pH 8.0), 50 mM sodium chloride. Following dialysis active AEP was separated from insoluble material by centrifugation and purified by size-exclusion chromatography

(HiLoad 16/600 Superdex 200) in 100 mM citric acid – sodium citrate buffer (pH 4.0) containing 50 mM sodium chloride. Protein was assessed for purity by SDS-PAGE.

HaAEP1 site-directed mutations were made following the Stratagene QuickChange protocol with the following primers: The N73A mutation was made with forward primer 5′-GTA GCA AAG GTT ATG GTG CTT ATC GTC ATC AGG CC-3′ and reverse primer 5′-GGC CTG ATG ACG ATA AGC ACC ATA ACC TTT GCT AC-3′; the N73D mutation with 5′-GTA GCA AAG GTT ATG GTG CTT ATC GTC ATC AGG CC-3′ and 5′-GCC TGA TGA CGA TAA TCA CCA TAA CCT TTG CTA C-3′; The D177A mutation with 5′-CTG TTT TAT AGC GCT CAT GGT GGT CCG G-3′ and 5′-CCG GAC CAC CAT GAG CGC TAT AAA ACA G-3′; the D177G mutation with 5′-CTG TTT TAT AGC GGC CAT GGT GGT CCG GG-3′ and 5′-CCC GGA CCA CCA TGG CCG CTA TAA AAC AG-3′; the H178A mutation with 5′-CTG TTT TAT AGC GAT GCT GGT GGT CCG GGT G-3′ and 5′-CAC CCG GAC CAC CAG CAT CGC TAT AAA ACA G-3′; the G185S mutation with 5′-GTC CGG GTG TTC TGA GTA TGC CGA ATG AAC-3′ and 5′-GTT CAT TCG GCA TAC TCA GAA CAC CCG GAC-3′; the C220S mutation with 5′-TGA TTT ATC TGG AAG CAT CTG AAA GCG GCA GCA T-3′ and 5′-ATG CTG CCG CTT TCA GAT GCT TCC AGA TAA ATC A-3′; and the E221K mutation with 5′-GAT TTA TCT GGA AGC ATG TAA GAG CGG CAG CAT TTT TGA AGG-3′ and 5′-CCT TCA AAA ATG CTG CCG CTC TTA CAT GCT TCC AGA TAA ATC-3′. Mutations were verified by sequencing and expressed as described for WT.

## Crystallization and data collection

Protein was concentrated to 10–15 mg/mL and crystal screening performed using the sitting-drop-vapor diffusion method with 80 μL of reservoir solution in 96-well Intelli-Plates at 16°C. Protein to mother-liquor ratios for the sitting drops were varied in each condition: 0.1:0.1, 0.1:0.2, and 0.2:0.1 μL. Crystals of active HaAEP1 were obtained in 0.1 M HEPES (pH 7.5), 1.4 M sodium citrate tribasic dihydrate. Single crystals were soaked in mother-liquor supplemented with 20% glycerol as a cryoprotectant prior to being flash-frozen and stored in liquid nitrogen. Data collection was performed at 100 K on the Australian MX2 (micro-focus) beamline using a wavelength of 0.9537 Å and diffraction data for crystals were collected to a resolution of 1.8 Å.

## Structural determination, refinement and model building

Diffraction data were processed using iMOSFLM and scaled with AIMLESS from the CCP4 program suite (*Battye et al., 2011*; *Winn et al., 2011*) in space group P3$_1$21 with unit cell dimensions $a = b = 77.03$, $c = 108.17$. A sequence alignment of HaAEP1 and human AEP1 was generated using ClustalO and used to create a search model of HaAEP1 based on the last common atom of human AEP (4FGU) using CHAINSAW. The structure of HaAEP1 was solved by molecular replacement with PHASER using this search model, followed by automatic building with ARP/WARP. Manual building and refinement was performed in iterative cycles with COOT and REFMAC5 using the CCP4 program suite. Structural analysis and validation were carried out with COOT and MolProbity (*Emsley and Cowtan, 2004*; *Chen et al., 2010*). Crystallographic data and refinement statistics are summarized in *Table 1* with Ramachandran plot values calculated from COOT. The peptide AAN modeled into the HaAEP1 active site was oriented into the active site based on the similar mode of cystatin binding to human AEP (4N6O)(*Dall et al., 2015*). The tetrahedral intermediate was evidenced from initial visualization in Fo-Fc electron density difference maps using a polder OMIT map as implemented in phenix.polder (*Liebschner et al., 2017*). Coordinates and structure factors were deposited into the Protein Data Bank (PDB) under accession code 6AZT. Tetrahedral complex of a three residue peptide ligand (AAN) bound to the active site Cys220 defined as CX9 in PDB file. Figures illustrating structures were generated using PyMol, electrostatic surface potentials were contoured at ±10 kT/e using an APBS PyMol plugin (*Dolinsky et al., 2007*; *Schrodinger, 2010*). Models of *C. ensiformis* and *C. ternatea* AEPs were generated using the I-TASSER server (*Roy et al., 2010*).

## Circular dichroism

Proteins purified in 50 mM Tris (pH 8.0) 300 mM imidazole were concentrated in an 30 kDa Amicon centrifugal filter and buffer exchanged with an excess of 10 mM sodium phosphate buffer (pH 8). Concentrations were checked by absorbance at 280 nm with a NanoDrop using the extinction coefficient of Pro-AEP and samples diluted to 0.1 mg/ml or 0.2 mg/ml. CD measurements were made in

triplicate using a Jasco J-810 CD spectrometer with a 0.1 cm quartz cuvette using a 1 nm bandwidth on a 0.1 mg/ml sample. CD wavelength spectra were collected from between 200–260 nm a rate of 1 nm/sec at 20°C. Melt curves were collected using the same bandwidth at 222 nm with temperature increasing at a rate of 1 °C/min from 20–95°C on a 0.2 mg/ml sample. WT HaAEP1 melting temperature was interpolated from melt curve using a sigmoidal four parameter logistic regression fit (GraphPad Prism, La Jolla, USA).

## Protein activity analysis

AEPs were activated by dialysis for four hours at room temperature in activation buffer (20 mM sodium acetate pH 4.0, 5 mM DTT, 100 mM sodium chloride, 1 mM EDTA) followed by a second dialysis into ligation buffer (20 mM MES pH 6.5, 0.5 mM DTT, 100 mM sodium chloride, 1 mM EDTA acid). Protein concentrations were determined by measuring absorbance at A280 using a NanoDrop (1 Abs = 1.0 mg/mL). For mass spectrometry analysis of the processing of SFTI(D14N)-GLDN or native SFTI-GLDN by WT HaAEP1, RcAEP, AtAEP2 and mutant HaAEPs, AEPs at a concentration of 40 µg/mL were incubated with 0.25 mM peptide with a diselenide bond and 25 mM native (i.e. disulfide) SFTI-1 as an internal standard in activity buffer (20 mM MES pH 6.5, 5 mM DTT, 1 mM EDTA). Reactions were carried out at 37°C for 16 hr. Three independent reactions were performed for each protein tested. The reactions were stopped by dilution 100-fold in 50% acetonitrile, 0.1% formic acid and spotted with an α-cyano-4-hydroxycinnamic acid matrix onto a plate for analysis by MALDI-MS. Quantification of peak area by MALDI-MS was calculated using the internal standard to normalize for ionization efficiency as described previously (*Bernath-Levin et al., 2015*). Briefly, for activity probe analysis 50 µL of AEP at 10 µg/mL was incubated with 1 µM of the BODIPY probe JOPD1 (*Lu et al., 2015*) at room temperature overnight and protected from light. The labeling reaction was stopped by the addition of 10 µL of 6x SDS-PAGE loading buffer containing β-mercaptoethanol and proteins separated using 4–12% Bis-Tris SDS-PAGE gels as previously described (*Lu et al., 2015*). Labeled proteins were visualized in gel with excitation and emission wavelengths of 532 and 580 nm using a Typhoon 9500 (GE Healthcare, Paramatta, Australia).

## Analysis of reversal of AEP activation

WT, D177G, N73A and E221K proteins were purified by affinity chromatography as described above and activated by overnight (4°C) dialysis at pH 4.0/5.5/6.5 in 100 mM citric acid - sodium citrate buffer containing 50 mM sodium chloride and 5 mM DTT, with pH adjusted by addition of 1.0 M Tris-HCl pH 8. Prior to pH-shift, sample aliquots were flash frozen and stored at −80°C for subsequent activity probe analysis and SDS-PAGE analysis. Remaining activated protein was then returned to the previous buffer but with pH adjusted to pH 8.0 by adding 1.0 M Tris-HCl pH 8.0, as previously described, and incubated overnight at 4°C (*Hara-Nishimura et al., 1998*). Following dialysis at pH 8.0 samples were flash frozen as described above. Proteins were then analyzed for activity, as described above, using the BODIPY activity based probe. Following in gel visualization of active protein gels were immediately fixed in Coomassie Blue stain for comparison of active and inactive protein.

## Acknowledgements

The authors thank Dan Tawfik for critical insights during project design, Hannes Ludewig for comments on the manuscript and Renier van der Hoorn for the BODIPY probe JOPD1. JSM was supported by an Australian Research Council (ARC) Future Fellowship (FT120100013). SGN and KVS were supported by the Australian Government's Research Training Program. JH and this work were supported by ARC grant DP160100107 to JSM and Dan Tawfik. This research was undertaken on the MX2 beamline at the Australian Synchrotron, part of the Australian Nuclear Science and Technology Organisation. HaAEP1 is deposited in PDB under accession 6AZT.

## Additional information

### Funding

| Funder | Grant reference number | Author |
| --- | --- | --- |
| Australian Research Council | FT120100013 | Joshua S Mylne |
| Australian Research Council | DP160100107 | Joshua S Mylne |

The funders had no role in study design, data collection and interpretation, or the decision to submit the work for publication.

### Author contributions

Joel Haywood, Conceptualization, Data curation, Formal analysis, Validation, Investigation, Visualization, Methodology, Writing—original draft, Writing—review and editing; Jason W Schmidberger, Data curation, Formal analysis, Validation, Investigation, Visualization, Methodology, Writing—review and editing; Amy M James, Data curation, Formal analysis, Validation, Investigation, Visualization, Writing—review and editing; Samuel G Nonis, Kirill V Sukhoverkov, Investigation, Visualization, Writing—review and editing; Mikael Elias, Formal analysis, Validation, Investigation, Visualization, Writing—review and editing; Charles S Bond, Formal analysis, Supervision, Validation, Visualization, Writing—review and editing; Joshua S Mylne, Conceptualization, Resources, Supervision, Funding acquisition, Validation, Visualization, Methodology, Writing—original draft, Project administration, Writing—review and editing

### Author ORCIDs

Joshua S Mylne (iD) http://orcid.org/0000-0003-4957-6388

### Decision letter and Author response

Decision letter https://doi.org/10.7554/eLife.32955.024
Author response https://doi.org/10.7554/eLife.32955.025

## Additional files

### Supplementary files

• Transparent reporting form
DOI: https://doi.org/10.7554/eLife.32955.020

### Major datasets

The following dataset was generated:

| Author(s) | Year | Dataset title | Dataset URL | Database, license, and accessibility information |
| --- | --- | --- | --- | --- |
| Bond CS | 2017 | Asparaginyl endopeptidase 1 bound to AAN peptide, a tetrahedral intermediate | https://www.rcsb.org/structure/6AZT | Publicly available at Protein Data Bank (accession no. 6AZT) |

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
