## [Decision Letter]

Thank you for choosing to send your work entitled "Structural basis of ribosomal peptide macrocyclization in plants" for consideration at *eLife*. Your article has been reviewed by three peer reviewers, and the evaluation has been overseen by a Reviewing Editor, Charles Craik, and a Senior Editor, Michael Marletta. The reviewers have discussed the reviews with one another and Charles Craik has drafted this decision to help you prepare a revised submission.

The current work presents the first structure of an "active" cysteine protease asparaginyl endopeptidase (AEP) that is capable of generating a macrocycle and visualizes a tetrahedral intermediate bound at the active site. While there have been structures of other AEPs, this work directly addresses the question of how AEPs actually produce macrocyclic products. Through mutagenesis and modulation of the pH, the hydrolytic vs. the transamination activity of the human homolog of AEP was dissected suggesting that deprotonation of the α amino group facilitates the macrocyclization of the covalently bound substrate at the thioacylenzyme intermediate. The work is of general interest to chemists and biotechnologists particularly due to the current interest in the macrocyclization process for its use in generating diverse cyclopeptide scaffolds that could be used as therapeutic leads.

The manuscript is in principle within the scope of *eLife*'s interest/mission. The reviewers raise valid points and a summary of our assessment follows:

Structural comments:

In light of the remarkable result of identifying a trapped tetrahedral intermediate, a precise justification of the presence of a tetrahedral intermediate in the active site instead of some other substrate or intermediate is justified in the text. What is it about the structural model that inspired the authors to model in a tetrahedral intermediate? Related to this, at the end of the subsection “HaAEP1 displays a tetrahedral intermediate in the active site”, Dall et al., 2015 is cited, but the evidence in the reference that "this tetrahedral intermediate state has not been described before." A more precise justification could clarify.

A map showing this result should be very clear. In place of the electron density maps in Figure 2 stereo view would improve the clarity of the trapped tetrahedral intermediate, and a simulated annealing omit map would be better than a 2|Fo|-|Fc| map which would make a more compelling case for alternate conformations with minimal model bias.

For part (B), although an omit map (|Fo|-|Fc|) is shown, the authors do not say whether this is a simulated annealing omit map. If it is not, check the simulated annealing omit map to ensure that model bias does not significantly influence the observed density. The atoms of C220, the conformer involved in the thioacylenzyme, should be omitted in the calculation.

The authors state that Brady et al. 1999, "…suggested that a local Gly amide backbone might polarize a water molecule in the absence of His to enable peptide hydrolysis, analysis of the HaAEP1 structure reveals Gly179 could potentially perform this role." There is some confusion since it is not clear how a hydrogen bond with a backbone NH group could polarize a water molecule and Brady et al. describe a glycine-hydrogen bonded water molecule that might serve as a proton donor. It should be clarified how the authors are referring to this work.

In Table 1, it is not meaningful to report thermal B factors to the hundredths place; rounding these to the tenths place, or simply rounded integer values, would be sufficient for a structure determined at 1.80 Å resolution.

Functional comments:

An appealing presentation of data on the kinetic rate constants and the substrate specificity of the enzyme is warranted. Kinetic constants should be included for at least the wild-type enzyme. Data from the literature or independently generated by the authors should be included since providing the kinetic competency of the enzyme will elevate interests from the biotechnological community.

The substrate selectivity of HaAEP1 should be addressed. What are its endogenous substrates and what is its biological role? Does it natively synthesize SFTI-1?

Provide clarity about the functional aspects of the enzyme for a broad audience, since the paper is currently challenging to follow without significant background reading. For example:

Although it is stated in the paper, it is not clear in the Introduction that HaAEP1 produces a mixture of linear and cyclic precursors until later in the Results. Is this thought to be an endogenous reaction that is relevant to its biological role? Is it known to be condition dependent?

The experiment exemplified by Figure 3 is not well explained in the text. What does the addition of seleno vs. wild type reveal? Is the quantitation appropriate? Are relative peak areas informative given the potential for different ionization efficiencies? See also the suggestion below regarding an additional figure for clarity.

Editorial comments:

AEPs are members of the ribosomally synthesized and post-translationally modified peptide (RiPP) class of natural products. This should be noted in the Introduction and referenced appropriately.

For the benefit of non-specialist readers, a figure should be considered that introduces different macrocyclic peptides that are products of various biosynthetic enzymes (i.e. SFTI, orbitides, cyanobactins, etc.). This figure could also include a diagram depicting the AEP reaction (linear peptide cleaved into macrocyclic peptide+linear tail).

"nucleophilic attack of the carbonyl carbon" should read "nucleophilic attack on the carbonyl carbon".

The fifth paragraph of the Discussion contains information that describe an essential result and should be rewritten for clarity and included in the Results section.

---

## [Author Response]

The manuscript is in principle within the scope of eLife's interest/mission. The reviewers raise valid points and a summary of our assessment follows:Structural comments:In light of the remarkable result of identifying a trapped tetrahedral intermediate, a precise justification of the presence of a tetrahedral intermediate in the active site instead of some other substrate or intermediate is justified in the text. What is it about the structural model that inspired the authors to model in a tetrahedral intermediate? Related to this, at the end of the subsection “HaAEP1 displays a tetrahedral intermediate in the active site”, Dall et al., 2015 is cited, but the evidence in the reference that "this tetrahedral intermediate state has not been described before." A more precise justification could clarify.A map showing this result should be very clear. In place of the electron density maps in Figure 2 stereo view would improve the clarity of the trapped tetrahedral intermediate, and a simulated annealing omit map would be better than a 2|Fo|-|Fc| map which would make a more compelling case for alternate conformations with minimal model bias.For part (B), although an omit map (|Fo|-|Fc|) is shown, the authors do not say whether this is a simulated annealing omit map. If it is not, check the simulated annealing omit map to ensure that model bias does not significantly influence the observed density. The atoms of C220, the conformer involved in the thioacylenzyme, should be omitted in the calculation.

To clarify our justification for modeling a trapped tetrahedral intermediate in the active site we have included a new figure supplement (Figure 3—figure supplement 2) illustrating our modeled tetrahedral intermediate with accompanying polder OMIT maps (F_obs –_ F_calc_). The figure supplement also illustrates our reasoning for modeling a tetrahedral intermediate over alternative substrates. Furthermore, as suggested we have introduced a cross-eyed stereo view of a polder OMIT map (F_obs –_ F_calc_) calculated in the absence of the shown overlaid AAN tetrahedral intermediate, contoured at 3σ level into Figure 3. Figure 3 has also been modified accordingly and the alternate conformations of the catalytic Cys and His are illustrated with a simulated annealing omit electron density map (2 F_obs_ – F_calc_) contoured at 1σ level.

We have also clarified our justification for the tetrahedral intermediate in the main text:

“The presence of this unexpected substrate in the HaAEP1 active site is likely to be a product of auto-activation. […] Although alternative conformations of the active site Cys in hAEP have been described previously, this tetrahedral intermediate state has not been described before (Dall et al. 2015).”

The Figure 3 legend text now reads:

Outstanding features of the HaAEP1 active site. (A) Catalytic residues with dual conformations illustrated in the HaAEP1 active site with simulated annealing omit electron density maps (2 F_obs_ – F_calc_) contoured at 1σ level. (B)Cross-eyed stereo view of polder OMIT map (F_obs –_ F_calc_) calculated in the absence of the shown overlaid AAN tetrahedral intermediate, contoured at 3σ level.

The Materials and methods text now reads:

“The peptide AAN modeled into the HaAEP1 active site was oriented into the active site based on the similar mode of cystatin binding to human AEP (4N6O)(Dall et al. 2015).[…] Coordinates and structure factors were deposited into the Protein Data Bank (PDB) under accession code 6AZT.”

The authors state that Brady et al. 1999, "…suggested that a local Gly amide backbone might polarize a water molecule in the absence of His to enable peptide hydrolysis, analysis of the HaAEP1 structure reveals Gly179 could potentially perform this role." There is some confusion since it is not clear how a hydrogen bond with a backbone NH group could polarize a water molecule and Brady et al. describe a glycine-hydrogen bonded water molecule that might serve as a proton donor. It should be clarified how the authors are referring to this work.

In order to clarify this point we have altered the text accordingly:

“Previous reports have also suggested that a local Gly amide backbone might facilitate catalysis via a transfer of a proton to the leaving group of the tetrahedral intermediate from a hydrogen bonded water molecule; analysis of the HaAEP1 structure reveals Gly179 could potentially perform this role in the absence of His (Brady et al., 1999).”

In Table 1, it is not meaningful to report thermal B factors to the hundredths place; rounding these to the tenths place, or simply rounded integer values, would be sufficient for a structure determined at 1.80 Å resolution.

The thermal B factors have been altered to the tenths place as suggested.

Functional comments:An appealing presentation of data on the kinetic rate constants and the substrate specificity of the enzyme is warranted. Kinetic constants should be included for at least the wild-type enzyme. Data from the literature or independently generated by the authors should be included since providing the kinetic competency of the enzyme will elevate interests from the biotechnological community.

Kinetic data for wild-type HaAEP1 enzyme has been included from the literature in the main text:

“HaAEP1 had previously been unable to create a macrocyclic product from SFTI(D14N)-GLDN, but had been shown to efficiently cleave it at a rate, k_cat_/K_m_ value of 610 M^-1^ S^-1^, similar to rates published for other AEPs (Bernath-Levin et al. 2015).”

The substrate selectivity of HaAEP1 should be addressed. What are its endogenous substrates and what is its biological role? Does it natively synthesize SFTI-1?Provide clarity about the functional aspects of the enzyme for a broad audience, since the paper is currently challenging to follow without significant background reading. For example:Although it is stated in the paper, it is not clear in the Introduction that HaAEP1 produces a mixture of linear and cyclic precursors until later in the Results. Is this thought to be an endogenous reaction that is relevant to its biological role? Is it known to be condition dependent?

The biological role and functional aspects of HaAEP1 have been addressed with new text in several areas of the Introduction to provide clarity for readers new to the field.

The text now reads:

“The recent discovery that evolutionarily distinct plant families have repeatedly recruited AEPs to catalyze the formation of ribosomally synthesized and post-translationally modified peptides (RiPPs), through the macrocyclization of linear precursor sequences, has caught the attention of drug designers keen to overcome the current inefficiencies in native chemical ligation that limit the therapeutic use of cyclic peptides (Pattabiraman and Bode 2011, Mylne et al. 2012, Arnison et al. 2013).”

And:

“Sunflower trypsin inhibitor-1 (SFTI-1) is a 14-residue, bicyclic peptide with a cyclic backbone and an internal disulfide bond (Luckett et al. 1999). […] Along with an adjacent albumin, SFTI-1 is post-translationally processed by AEP from within a unique seed storage albumin precursor called Preproalbumin with SFTI-1 (PawS1) (Mylne et al. 2011).”

And:

“Notably, this reaction proceeds in competition with nucleophilic attack upon the thioacyl intermediate by any nearby water molecules which would produce hydrolyzed, acyclic-SFTI. […] Evidence for AEP-mediated and hydrolysis-independent transpeptidation was demonstrated through the exclusion of a heavy atom O^18^ in the cyclic SFTI-1 product from an in vitro jack bean AEP (CeAEP1) catalyzed reaction (Bernath-Levin et al. 2015).”

The experiment exemplified by Figure 3 is not well explained in the text. What does the addition of seleno vs. wild type reveal? Is the quantitation appropriate? Are relative peak areas informative given the potential for different ionization efficiencies? See also the suggestion below regarding an additional figure for clarity.

Several areas of the main text have been modified accordingly to clarify Figure 3 and Figure 3:

“Incubation of WT HaAEP1 with a fluorophore labeled (BODIPY) activity-based probe (Lu et al. 2015) illustrated its heterogeneity in size following activation at pH 4 and incubation with substrate at pH 6.5, which has previously been observed for AEP proteins both in vivo and in vitro and speculated to be the result of processing of non-glycosylated forms (Zhao et al. 2014) (Figure 4). […] As expected, mutation of the second residue of the catalytic dyad (H178A) also results in a drastic reduction in activity, based on SFTI(D14N)-GLDN processing and activity based BODIPY probe results, confirming the significance of C220 and H178 in AEP activity (Figure 4, Figure 4—figure supplement 1).”

And:

“The mutation E221K, which has previously been shown to increase endopeptidase activity in hAEP, resulted in a loss of cyclic product as shown by an absence of a peak of mass 1608 (Figure 4, Figure 4—figure supplement 1). Mutation of N73A leads to a higher ratio of cyclic to acyclic product as shown by an increased peak area relative to WT of mass 1608 and a reduced acyclic product peak of mass 1626.”

The previous Figure 3 is now Figure 4 and the legend reads:

“…(C) Quantitation of peak areas from B. Peak areas of mass 1608 – cyclic SFTI(D14N), mass 1626 – acyclic-SFTI(D14N) and mass 2025 – unprocessed seleno-Cys modified SFTI(D14N)-GLDN substrate, were normalized for ionization efficiency using an internal standard mass 1515 – native SFTI-1. […] Error bars illustrate standard deviation n=3 (D177A n=2) technical replicates.”

Editorial comments:AEPs are members of the ribosomally synthesized and post-translationally modified peptide (RiPP) class of natural products. This should be noted in the Introduction and referenced appropriately.For the benefit of non-specialist readers, a figure should be considered that introduces different macrocyclic peptides that are products of various biosynthetic enzymes (i.e. SFTI, orbitides, cyanobactins, etc.). This figure could also include a diagram depicting the AEP reaction (linear peptide cleaved into macrocyclic peptide+linear tail).

The text and references have been edited accordingly and we have added a new figure that introduces enzymes that catalyse the macrocyclization of a variety of cyclic RiPPs (Figure 1):

“The recent discovery that evolutionarily distinct plant families have repeatedly recruited AEPs to catalyze the formation of ribosomally synthesized and post-translationally modified peptides (RiPPs), through the macrocyclization of linear precursor sequences, has caught the attention of drug designers keen to overcome the current inefficiencies in native chemical ligation that limit the therapeutic use of cyclic peptides (Pattabiraman and Bode 2011, Mylne et al. 2012, Arnison et al. 2013). […] Moreover, as computational techniques for the discovery of RiPPs improve and the number of cyclic peptides described continues to expand, an ever-wider array of scaffolds might be exploited to tailor molecules to specific drug targets (Bhardwaj et al. 2016, Truman 2016, Hetrick and van der Donk 2017) (Figure 1).”

The new Figure 1 legend reads:

“Examples of enzyme catalyzed formation of cyclic RiPPs. Cyclic RiPPs that are enzyme catalyzed from linear peptide precursors are commonly flanked by an N-terminus leader sequence and a C-terminus follower sequence prior to cyclization. […] These examples also illustrate a range of cyclic peptide scaffolds that may have potential therapeutic applications.”

"nucleophilic attack of the carbonyl carbon" should read "nucleophilic attack on the carbonyl carbon".

The text has been edited as suggested.

The fifth paragraph of the Discussion contains information that describe an essential result and should be rewritten for clarity and included in the Results section.

The text describing the result has been rewritten for clarity and broken in two, some has been moved to the Results section and part remains in the Discussion, with elaboration:

The Results now reads:

“AEPs have an intrinsic ability to perform peptide macrocyclization

Given the sequence similarity between AEPs and the conservation of residues involved in catalysis we hypothesized that the ability to macrocyclize peptides might be inherent to AEPs. […] These findings further supports our hypothesis that the structural features of AEPs described above have allowed for the convergence upon AEP for peptide macrocyclization reactions.”

The Discussion now reads:

“The finding that AEPs from species that lack cyclic peptides may be coaxed into performing peptide macrocyclization of a linear peptide under favorable conditions significantly expands the potential use of AEPs for the production of cyclic peptides (Figure 6). […] These differences include shorter sidechains in the β_I_ sheet and β5-β_IV_ loops that may reduce steric hindrance on a peptides N-terminus during attack on a thioacyl intermediate (HaAEP1: P181, Q245, N247 OaAEP1: A178, T242, S244 Butelase-1: A168, G232, S234) and differences in the β_IV-_β_V_ region that could generate substrate specificity (Figure 5—figure supplement 1).